# Comparative Effects of Different Light Sources on the Production of Key Secondary Metabolites in Plants In Vitro Cultures

**DOI:** 10.3390/plants10081521

**Published:** 2021-07-26

**Authors:** Mariam Hashim, Bushra Ahmad, Samantha Drouet, Christophe Hano, Bilal Haider Abbasi, Sumaira Anjum

**Affiliations:** 1Department of Biotechnology, Kinnaird College for Women, Jail Road, Lahore 54000, Pakistan; mariamhashim07@gmail.com; 2Shaheed Benazir Bhutto Women University, Peshawar 25000, Pakistan; bushraahmad@sbbwu.edu.pk; 3Laboratoire de Biologie des Ligneux et des Grandes Cultures, INRAE USC1328, Eure & Loir Campus, University of Orleans, 28000 Chartres, France; samantha.drouet@univ-orleans.fr (S.D.); hano@univ-orleans.fr (C.H.); 4Department of Biotechnology, Quaid-i-Azam University, Islamabad 15320, Pakistan; bhabbasi@qau.edu.pk

**Keywords:** LED light, fluorescent light, UV light, elicitation, plant secondary metabolites, plant in vitro cultures

## Abstract

Plant secondary metabolites are known to have a variety of biological activities beneficial to human health. They are becoming more popular as a result of their unique features and account for a major portion of the pharmacological industry. However, obtaining secondary metabolites directly from wild plants has substantial drawbacks, such as taking a long time, posing a risk of species extinction owing to over-exploitation, and producing a limited quantity. Thus, there is a paradigm shift towards the employment of plant tissue culture techniques for the production of key secondary metabolites in vitro. Elicitation appears to be a viable method for increasing phytochemical content and improving the quality of medicinal plants and fruits and vegetables. In vitro culture elicitation activates the plant’s defense response and increases the synthesis of secondary metabolites in larger proportions, which are helpful for therapeutic purposes. In this respect, light has emerged as a unique and efficient elicitor for enhancing the in vitro production of pharmacologically important secondary metabolites. Various types of light (UV, fluorescent, and LEDs) have been found as elicitors of secondary metabolites, which are described in this review.

## 1. Introduction

Plants are complex species and have gained importance due to their nutritional and pharmaceutical values. Apart from the production of primary metabolites such as carbohydrates, lipids, and proteins that plants need for their growth and development, low molecular weight organic compounds involved in defense against stress conditions called secondary metabolites are also synthesized by higher plants [1]. Secondary metabolites are involved in the production of pharmaceuticals, industrially important biochemicals, food additives, and flavors [2]. The production of secondary metabolites in the wild is limited to some re-gional and environmental constraints, which limit the production of compounds commercially [3]. Traditional cultivation of certain types of plants is often difficult and may take several years for their growth [4].Recent trends have focused on developing in vitro culture techniques as a convenient alternative to cope with the demand for medicinal plants, as more than 60% of anti-cancer drugs are manufactured directly or indirectly from plants [5,6]. In vitro cultures are an efficient means of production of biomass, leading to rapid growth and consistent metabolite productivity [7]. In addition, elicitation has proved beneficial in the production of in vitro cultures [2]. Usually, under stress conditions or environment variability, the output of these compounds is improved to cope with se-vere stress effects. For scaling up the development of these phytochemicals, in vitro techniques can prove beneficial.

Elicitation, where certain pathways are activated by introducing agents (elicitors) for triggering a plant’s defense mechanisms, is amongst the most relevant and effec-tive techniques [3,8]. To increase the production of secondary metabolites, many biotic and abiotic elicitors are used. Light is, however, an influential abiotic elicitor that af-fects the growth, development, and morphogenesis in plants [9,10]. Light also plays a critical role in controlling primary and secondary metabolism in order to achieve op-timum growth in plants [11,12,13]. Light stress has been designed to increase secondary metabolite production from various in vitro cultures of medicinally im-portant plants [12,14]. The signaling, regulatory, and metabolic mechanisms involved in eliciting secondary metabolites, as well as the mechanism of light precipitation, are not thoroughly characterized in the literature. However, it is reasonable to speculate that oxidative stress, in addition to other mechanisms, plays a significant role in light perception and signaling. Oxidative damage produced as a result of environmental stress leads to the production of highly reactive free radicals that halts the growth and development of plants [15,16,17]. To counteract the effect of these radicals, plants have natural antioxi-dant defense mechanisms that are involved in producing a wide range of secondary metabolites [18,19,20].

The improved production of various valuable secondary metabolites through light elicitation has unlocked a new area of research that could have significant economic benefits for the pharmaceutical and nutraceutical industry. To date, different sources of light such as ultraviolet (UV), light-emitting diodes (LED) and fluorescent lights have been reported as efficient elicitors of pharmacologically important secondary metabolites, as summarized in Figure 1 [21,22,23]. These light sources have been used either alone or in combination with each other in order to maximize the production of valuable metabolites in in vitro cultures of plants. In this review, in-depth literature on the role of light as an elicitor of valuable secondary metabolites has been critically reviewed. Furthermore, the mechanistic aspects of various sources of light as elicitors of secondary metabolites through activation/regulation of various genes are also discussed.

## 2. Light as an Elicitor

In vitro cultures of several plant species have been documented to elicit secondary metabolites using a variety of light sources in the past. These sources have been classified into three main categories in this review as UV lights, LED lights, and fluorescent lights, compared and discussed herein in detail.

### 2.1. UV Lights

A significant abiotic elicitor used in the past to boost the production of secondary metabolites in a variety of plant cultures is UV [24]. The wavelength of UV (400–200 nm) accounts for only a small por-tion of the solar radiation that reaches the earth’s surface, yet it has a significant bio-logical impact on living species, including plants. UV radiation is categorized into three parts: UV-A (320 to 390 nm), UV-B (280 to 320 nm), and UV-C (280 nm and be-low). The intensity of UV-A is greater than UV-B, but this difference is not biologically significant [25]. UV-B photons are perhaps the most intense wave that hits the surface of the earth, and even tiny changes in their quantity can have a huge impact on vital processes and properties at all scales, from species to ecosystems [26]. In addition to the rest of the UV groups, UV-C radiation has also proved to be the most effective in stimulating the production of plant secondary metabolites such as phenolics, alkaloids, or flavonoids [27,28,29] (Table 1). 

#### 2.1.1. UV-A and UV-B

Since UV light acts as an elicitor, it is basically involved in activating the defense mechanisms of plants which in turn produce secondary metabolites useful for humans for therapeutic purposes as they are not required by plants for their growth [28]. UV-A light can act as a potential elicitor to stimulate the production of secondary metabolites in plants grown under controlled conditions and/or in vitro cultures (Table 1).

Flavonols, also known as 3-hydroxylavones, are the most common flavonoids found in food. They are structurally similar to flavones, but they differ in that they have a hydroxyl group at the 3-position on the C-ring, while flavones have a ketone group with an unsaturated carbon–carbon bond [44]. Cynaroside (luteo-lin-7-glucoside), a flavone, is used for a variety of medical purposes; it may protect heart cells from apoptosis caused by reactive oxygen species (ROS). Cynaroside also reduces kidney damage caused by the chemotherapeutic drug cisplatin, which is used to treat cancer [45]. The production of cynaroside has been increased in in vitro culture of Capsicum annum (aka bell pepper plant) by elicitation of UVA/B light for a period of 16 days [30]. 

UV-B has also been reported as a potential elicitor candidate to induce various changes in the metabolism of plants [46]. This leads to the activation of plant protec-tion mechanism by the formation of secondary metabolites such as alkaloids and fla-vonoids [47,48]. Physical responses in plant tissues, such as increased amounts of spe-cific phenolics, make plants more resistant to UV-B radiation than other species, and improved levels of pigmentation are induced [49,50]. Plant secondary metabolites containing polyphenolic structure, i.e., flavonoids, can be found in a wide range of foods, including fruits and vegetables. They have antioxidant and biochemical properties that can help with disorders like cancer, Alzheimer, atherosclerosis, and many others [51,52,53]. In a recent study, elicitation of *Nymphoides humboldtiana* with UV-B reported the production of pharmaceutically important flavonoids such as phloroglucinol, chlorogenic acid, epicatechin, quercetin, and ferulic acid [31]. Similarly, shoots of *Alternanthera* species, i.e., *A. sessilis* and *A. brasiliana*, on elicitation with UV-B for 8 h showed a 51% and 62% increase in flavonoid content, respectively, in comparison with control [32]. Likewise, an increase in the production of flavonoids was also reported in *Capsicum annum* L. with the elicitation of UV- B [33]. Under UV-B exposure, the production of flavonols, particularly quercetin, kaempferol, and isorhamnetin, were also vastly improved in *Ginkgo biloba* leaves [34]. Vinblastine and vincristine are chemotherapy drugs that are made by linking the alkaloids catharanthine and vindoline and are used to treat a variety of cancers [54]. The impact of UV-B on cell suspension cultures of the *Catharanthus roseus* plant was investigated in which the production of these important alkaloids, catharanthine and vindoline, was improved to 3 and 12 fold, respectively [35]. *Ocimum basilicum*, also known as sweet basil, possesses a wide range of potent activities due to the presence of precious secondary metabolites. It is used in traditional medicine as a result of its bioactive dary metabolites. It is used in traditional medicine as a result of its bioactive metabo-lites [55,56]. Elicitation with UV-B irradiation at an intensity of 224 μmol m^−2^ s^−1^ dramatically raised the production of anthocyanin, phenolics, and flavonoids in leaves of *Ocimum basilicum* [21].

When aerobic or photosynthetic metabolism is disrupted by various environmental stresses, phenolic compounds can play an important role in plant development by functioning as protective substances and signal molecules in plants, as well as safeguarding them from ROS. According to several studies, whenever a plant is infected with a disease, phenolic compounds are produced in response to that infection [57]. Most phenolics were accumulated when UV-B radiation was applied at a rate of 20 W/cm^2^ to wheat seedlings. On day 4, total phenolics, DPPH, and ABTS levels increased by 26.3, 25.1, and 12.0%, respectively, as compared to un-irradiated wheat seedlings [37]. Likewise, in another study, seeds of *Ocimum basilicum* on irradiation of UV-B for three days showed enhanced production of phenolics [36].

#### 2.1.2. UV-C 

UV-C region of the spectrum includes wavelengths below 280 nm. These intense wavelengths are absorbed by ozone and do not reach the earth’s surface [58]. Thus, the application of UV-C on various plant cultures using artificial lamps may be a promising strategy in raising the production of valuable secondary metabolites. Trans-resveratrol (TR) and resveratrol are non-flavonoid phenolics of stilbenes and are classified under biologically active isomers found majorly in red grapes and berries. They are mainly employed in pharmaceutical industries due to their anti-carcinogenic, anti-diabetic, anti-acne, antioxidant and anti-inflammatory activities. These properties make them a promising candidate for novel drugs [59]. The elicited callus culture of *Vitis vinifera* L. showed 8–10 times higher production of resveratrol levels in the 48 h after UV-C application than in the first 24 h. The highest resveratrol accumulation was recorded to be 62.23 µg/g of FW in a 12-day old culture [38]. Likewise, in another study, calli that had been exposed to UV-C for 5 min showed higher content of TR (8.43 µg g^−1^) when harvested after 24 h. Catechin accumulation (8.89 mg/g) was also higher in calli exposed to UV-C for 10 min when harvested after 48 h. In conclusion, UV-C radiation aided the accumulation of secondary metabolites in the calli of the Okuzgozu grape cultivar [39].

Polyphenols found in *Lepidium sativum* L. have a diverse variety of medical and pharmaceutical purposes. Essential oils are also abundant in this therapeutic plant and exhibit exceptional anti-cancer properties in animal models as well as in various cell lines [60,61,62]. UV-C was irradiated on the callus of *L. sativum* for varying periods of time and different concentrations of melatonin. Phytochemical investigations revealed the production of three significant compounds: chlorogenic acid, kaempferol, and quercetin in callus culture. Cultures treated with melatonin (20 µM) showed three times higher production (36.36 mg/g DW), whereas cultures exposed to UV-C (60 min) showed a 2.5-fold increase (32.33 mg/g DW) in production in comparison with control (13.94 mg/g DW). This study compares both the elicitors and their effects on the initiation of physiological pathways in *L. sativum* for the formation of secondary metabolites [40].

*Linum usitatissimum* L. is considered a functional food and is widely used as oilseed crops in Europe [63,64]. It is a nutrient-dense diet that can be used as a substitute for food because it contains all dietary components [65,66]. When lignans and neolignans were quantified using reverse phase-high performance liquid chromatography (RP-HPLC), cultures of *L. usitatissimum* exposed to UV-C + photoperiod (16 h light/8 h dark) showed the presence of secoisolariciresinol diglucoside (SDG), dehydrodiconiferyl alcohol glucoside (DCG), guaiacylglycerol-β-coniferyl alcohol ether glucoside (GGCG), and lariciresinol diglucoside (LDG). UV-C radiation of 3.6 kJ/m^2^ resulted in a higher accumulation of SDG, LDG, and GGCG by 1.86-fold, followed by 2.25-fold and 1.33-fold in cell cultures grown under UV + photoperiod. Furthermore, the accumulation of DCG was raised by 1.36-fold in cell cultures grown under UV + dark under an intensity of 1.8 kJ/m^2^. Moreover, total phenolics, flavonoids, and antioxidants were also enhanced by 2.82-fold, 2.94-fold, and 1.04-fold, respectively, in cell cultures maintained under UV + photoperiod at a dosage of 3.6 kJ/m^2^ of UV-C radiation. These findings broadened the scope of what can be done in *L. usitatasimum* cell cultures to produce biologically active lignans and neolignans [41].

*Ocimum basilicum* L. var purpurascens (Lamiaceae) has gained popularity due to its ornamental and aromatic qualities as well as the presence of important volatile secondary metabolites such as rosmarinic acid, flavonoids, and anthocyanin s [67]. In a recent report, the effect of UV-C on the synthesis of phenylpropanoid metabolites was examined in in vitro culture of *Ocimum basilicum* L. When compared to the control, UV-C (10 min) exposure resulted in a 2.3-fold increase in rosmarinic acid (134.5 mg/g DW). Chichoric acid (51.52 mg/g DW) and anthocyanin (cyanide 0.50 mg/g DW) were about 4.1-fold greater after 50 min of UV-C exposure, whereas peonidin was 2.7-fold higher. Overall, UV-C proved to be efficient elicitors, as there was a positive association between induced phenolic compound synthesis and antioxidant activity of basil callus extracts [42]. Similarly, in another study formation of secondary metabolites in *Echinacea purpurea* callus and cell suspension cultures were observed after 24, 48, and 72 h of elicitation. Callus and cell suspension cultures were exposed to UV-C radiation for 15, 30, and 60 min, respectively. UV-C irradiation for 60 min was the most successful in promoting the aggregation of most secondary metabolites with varying incubation times depending on tissue culture meth ods [43]. This suggests that UV-C could be a potential elicitor of secondary metabolite synthesis in plants in vitro cultures.

### 2.2. Fluorescent Lights 

When plants are grown under sole-source electric lighting, lamp spectral customization can be an approach for achieving desired plant characteristics [68]. Fluorescent lights are a major source of light energy for stimulating the production of secondary metabolites in in vitro cultures [9]. Previous studies have confirmed that, depending on plant species, light quality has a direct impact on morphological and physiological responses [9,69]. For the management of controlled-environment agriculture facilities, it is vital to conserve electrical energy expenditures. In this regard, fluorescent lamps are considered much cheaper than LEDs [70]. They are particularly appealing for a variety of applications due to their high efficiency, outstanding color rendering, and extended life [71]. As a result, their use could be beneficial in triggering various in vitro cultures for increased secondary metabolite production (Table 2). 

In a study, enhanced production of secondary metabolites and biomass accumulation was observed in *Stevia rebaudiana* callus cultures on exposure to different fluorescent spectral lights. Maximum capability in enhancing total phenolic content (102.32 µg/g of DW), total antioxidant potential (11.63 µg/g DW), and total flavonoids content (22.07 µg/g DW) was shown under blue light [22]. Similarly, blue light increased total phenolic (23.9 mg/g DW) and flavonoids content (1.65 mg/g DW) in callus cultures of *Prunella vulgaris* [14]. Likewise, in another study, blue light promoted aggregation of metabolites to the greatest degree in shoot culture of *Scutellaria lateriflora*. The major flavonoids that showed the highest concentrations were baicalin, verbascoside, glucuronides, and wogonoside. Their levels were 1.49, 1.86, 1.54, and 2.05 times higher than the control under white light, respectiv ely [72]. Based on the findings of the above investigations, it can be hypothesized that blue light can operate as a potential elicitor in diverse in vitro cultures, promoting the formation of secondary metabolites.

Hypericin and pseudohypericin, which are derivates of naphtodianthrones, are structurally similar phenolic compounds that have gained importance commercially due to their unique activities [76]. However, conventional cultivation of these metabolites is unable to fulfill the fierce competition in the pharmaceutical industry in terms of both quantity and quality. Therefore, elicitation of root culture of *Hypericum perforatum* with red light showed the highest production of total hypericins (i.e., hyperin + pseudohypericin) (9.61 ± 0.3 µg/g), whereas the lowest on exposure to fluorescent light (7.12 ± 0.26 µg/g). Roots grown under red light also showed the highest content of flavonoids (41.17 ± 7.21 mg/g). In this study, the impact of blue light was also evaluated on the production of key secondary metabolites. A considerable increase in the production of metabolites was observed after one week of exposure; however, an inhibitory effect was detected after five weeks of incubation. Results also showed that the production of hypericin and total phenolic content were increased to 52% and 26%, respectively, in root culture after one-week exposure to blue ligh t [73]. 

*Hyptis marrubioides* has been traditionally used to cure infections related to gastrointestines, cramps, discomfort, and skin infections [77] in many regions. Effect of various fluorescent lights was observed on the production of important phenolic compounds in seed culture of *H. marrubioides*. White (0.308 mg/g of DW) and blue (0.298 mg/g of DW) light was shown to accumulate the highest amount of rutin, while red light improved plant development and increased dry weight and leaf number in in vitro-cultivated seeds of *H. marrubioides* [74]. Ginsenosides (saponins) found in the root extract of *Panax ginseng* are the most active components known to exhibit immunomodulatory properties and provide protection against heart and liver diseases [78]. Rb and Rg are formed from the structures of 20(S) protopanaxadiol and 20(S) protopanaxatriol, and are two important groups of ginsenosides [79]. The Rg group of ginsenosides (5.3–0.1 mg/g DW) accumulated more than the Rb group (3.7–0.7 mg/g DW) in hairy roots grown under fluorescent light. These findings imply that growing hairy roots in dim or bright settings can affect the Rb and Rg ginsenoside productions in in vitro cultures [75].

*Artemisia absinthium* L., often known as “Wormwood”, is referred to as a “universal treatment for all ailments” due to its therapeutic medical characteristics [80,81]. This plant has traditionally been used to treat diarrhea, cough, and common cold due [82] to its insecticidal, bitter [83,84], vermifuge, trematocidal [85], diuretic, and antispasmodic properties [86]. Total phenolics, total flavonoids, and antioxidant activity were found to be more supported by the green spectrum grown for three weeks under a photosynthetic photon flux density of 40–50 µmol m^−2^ s^−1^ in callus culture of *A. absinthium* [9]. As a result of the preceding studies, it is concluded that light elicitation (fluorescent, blue, red, green, and white light) has a positive influence and that different light regimes can aid in optimizing plant growth and developmental changes for the formation of commercially significant secondary metabolites in vitro.

### 2.3. LED Lights 

The amount of light a plant receives has a big impact on its development, growth, and production. Agriculture employs traditional artificial light sources such as high-pressure sodium lamps (HPSLs), metal-halide lamps (MHLs), and fluorescent lamps (FLs) to provide a controlled atmosphere. Fluorescent lamps have risen in favor among these. However, the wavelengths of these lightning sources span from 350 to 750 nm, and for plant growth and development, it is considered of low quality. They possess a limited lifetime of activity and a low photosynthetic flux, limiting their use in plant illumination systems that require a large agricultural production. LEDs technology has been able to be employed in a rising variety of new sectors, including plant growth and development, due to the implementation of new types of semiconductor materials. As a substitute to conventional lighting systems, LEDs have proved to be a smarter source of artificial lighting to provide controlled conditions in agriculture and in vitro systems. Since LEDs can emit over specified spectral areas, they may be utilized to manage the amount of photosynthetically active and photomorphogenic radiation required for plant growth and development. Matching LED wavelengths to photoreceptors in plants can allow for optimal output while also altering plant shape and metabolism. As a result, these solid-state light sources can be used in developing lighting lamps for sustainable production and photo-morphogenesis research [87].

Plant growth, production, and secondary metabolism are all influenced by light in general and light quality in particular. LEDs exist in various colors, i.e., white, blue, green, red, yellow, violet, and far-red. Many scientists believe that red (600–700 nm) and blue (400–500 nm) light are more crucial for stimulating photosynthesis than other light wavelengths since they have the highest photosynthetic photon efficacy values. Green light (500–600 nm) can, on the other hand, penetrate deep inside the leaf due to its high transmittance and reflectance [88]. LED lights have been widely used to elicit the production of key secondary metabolites in various plant culture systems (Table 3).

#### 2.3.1. Blue LED

Traditional lighting systems require color filters, but LED lighting systems may create the light of any desired color without them. LEDs are versatile enough to produce only the type of light that plants require. Plants require specific spectrums or colors for distinct morphogenic responses, and LED systems can be fine-tuned to provide only those. Blue light controls stomatal opening and transpiration, as well as preventing “red light syndrome” [103]. Thus, blue LEDs can play an efficient role in stimulating the production of pharmacologically active secondary metabolites, as demonstrated in Table 3.

Tomatoes (*Solanum lycopersicum* L.) are the world’s seventh most produced crop species and one of the year-round value crops grown in greenhouses. Tomato fruits are nutritionally dense because they include vital nutrients as well as phytochemicals that promote health [104]. LED was used as a source of artificial light to evaluate its effect on *S. lycopersicum* L. ‘Cuty’ (tomato seedlings). Results showed raised production of phenolics, flavonoids, and antioxidants under blue light when compared to control. This concludes that manipulating light quality using LED’s could stimulate the production of bioactive compounds and antioxidants [23].

The use of nutritional or therapeutic plant-based natural substances to treat disease has become a novel paradigm in clinical science. Flavonoids, particularly phenolic compounds, can be found in nearly all plants. Antioxidant, anti-cancer, anti-diabetic, and cardiovascular effects are observed in phenolics and flavonoids [105]. A study reported the highest concentrations of soluble protein and flavonoid in lettuce when exposed to blue light [89]. Another study found that leaves of synseed grown under blue LED had higher amounts of chlorophyll a, flavonoids, phenolics, and carotenoids than those grown under fluorescent or blue-red LED. Anti-oxidant activity was similarly boosted in synseed-derived seedlings grown in blue LED light [90]. Likewise, the overall polyphenol content in the blue LED treatment was also considerably higher than in the control treatment in *Anoectochilus roxburghii* [91,101]. Similarly, both blue and blue–violet light supplements boosted phenolic acid production in *Ocimum basilicum*, while in *Eruca sativa*, higher flavonoid synthesis was seen in response to both light supplements, but greater production was observed under blue– violet [92]. The above study concludes the potential of using blue LED as a promising elicitor in raising the production dramatically. 

#### 2.3.2. Red LED 

Red LEDs has been widely employed as an alternative source of illumination for in vitro survival and improved production of metabolites in medicinally important plants. Red light is considered the most efficient spectral light in driving photosynthesis, and many studies reported it as an efficient elicitor in elevating the production of pharmacologically active constituents in in vitro systems of various plants [106]. *Myrtus communis* has long been used in medicine and possesses therapeutic characteristics [107,108]. *M. communis* (leaves, flowers, and fruits) contain various components that are critical to the pharmaceutical, food, liqueur, and cosmetic industries; hence, large-scale manufacturing of this plant is necessary [109]. *M. communis* leaf extracts are enriched sources of phenolic acids and flavonoids. In a recent study, the impact of red LED was evaluated on in vitro culture of *M. communis*. Results showed that out of all flavonoids, myricetin (347.02–1118.69 mg 100 g^−1^ DW) was the main constituent with the highest concentration and 6-benzyladenine (BA) level, whereas gallic acid had the highest concentrations of the phenolic acids (95.58 mg 100 g^−1^ DW on average) at a photosynthetic photon flux density of 35 µmol m^−2^ s^−1^ [93]. 

Milk thistle (*Silybum marianum* L.) is a well-known hepato-protective medicinal herb that has been extensively researched. The effects of various LEDs lights were investigated, and it was discovered that red light greatly increased phenolics, flavonoids, and superoxide dismutase (SOD) activities in this plant. Under red light, HPLC analysis revealed a substantially doubled total silymarin concentration (18.67 mg g^−1^ DW) when compared to the control (9.17 mg g^−1^ DW). When exposed to red light, the levels of isosilychristin, silybin A, silybin B, silychristin, and silydianin were found to be highest. This demonstrates that the quality of light has a significant impact on the callus culture of *S. marianum* morphological and biochemical properties [94].

#### 2.3.3. Blue and Red LED in Combination

Several studies have reported the use of blue and red LEDs in combination on one or more plants simultaneously. Many reports suggested that the use of different LEDs in combination can enhance the production of bioactive compounds to several folds. Thus, their combination can be employed in eliciting various cultures that have gained importance pharmacologically [95,97,98]. Since ancient times, root extracts of *Rehmannia glutinosa* Libosch (Chinese foxglove) have been used to treat many disorders such as anemia and hypertension [96]. The therapeutic use of *R. glutinosa* roots is well known due to the presence of bioactive compounds such as catalpol, aucubin, and rehmaglutin [96,110]. In a study, plant development as well as phytochemicals with both defensive and prospective medicinal characteristics, such as phenolics and flavonoids, were influenced by spectral characteristics in both leaf and root tissues of *R. glutinosa*. On irradiation with blue LED, leaf extracts were found to have the highest total phenol concentration (35 ± 0.05 µg GAE/mg) as compared to root extracts. On the other hand, red LED exposure also increased the overall flavonoid content of the leaf extract by 33.6% and the root extracts by 61.7%. As a result, blue and red LEDs may be the most enticing light sources for *R. glutinosa* proliferation in in vitro conditions [95].

Bioactive compounds like phenolics and flavonoids are of significant importance in therapeutic plants because they can act as free radical scavengers [111]. The effects of red and blue LEDs on the accumulation of phenolic and flavonoids in *Ocimum basilicum* callus cultures were investigated in a recent study. Rosmarinic acid and eugenol were considerably enhanced under blue light (2.46 and 2.25 times greater than control). While under red light, the highest amounts of cyanidin (0.1216 mg/g DW) and peonidin (0.127 mg/g DW) were detected. Chicoric acid accumulation was about 4.52 times higher than callus grown under the control of continuous white light (81.40 mg/g DW) [97]. This suggests that employing LEDs to accelerate the production of physiologically active chemicals in vitro could be beneficial.

Fruits, nuts, medicinal herbs, and vegetables all contain phenylpropanoid, a type of secondary metabolite that is involved in reducing the risk of diabetes and heart disease through inhibiting carcinogenesis [112,113]. Plants are protected from bacteria by the antimicrobial activities of some phenylpropanoids during their interactions [114]. The generation of phenylpropanoid metabolites was examined using LEDs as an elicitor in wheat sprouts, and qRT-PCR and HPLC results revealed that white light (380 nm) was the best wavelength for epicatechin biosynthesis in wheat sprouts. Blue light (470 nm) was involved in the increased accumulation of gallic acid and quercetin, whereas red light (660 nm) increased the aggregation of ferulic acid on the 8th day and p-coumaric acid on the 12th day [98]. Similarly, flavonoids and anthocyanins, which also belong to a class of phenylpropanoids, were also enhanced in buckwheat, *Fagopyrum* sp. under red + blue (combination), and blue LED, respectively [99]. Likewise, phenolics and flavonoids were enhanced in seeds of *Ocimum basilicum* under red–blue (2R:1B) LED. In addition, the highest flavonoid content of 16.79 mmol/g FW was also achieved for protocorm-like bodies pre-treated with white LED for more than three cultures cycles under blue–red (1:1) LED [101]. Thus, red and blue LED ratios can be altered to generate improved growth and phenolic content in both red and green basil microgreens as a practical technique for generating superior quality foods [100]. Furthermore, in the callus culture of *Cnidium officinale*, mixed (red–blue) LED illumination enhanced the synthesis of phenolics and flavonoids [10]. In another study, in vitro generated *Eclipta alba* callus culture was subjected to multispectral lighting under regulated aseptic conditions. Results showed that the red light enhanced the production of phenolics (57.8 mg/g) and flavonoids (11.1 mg/g). whereas, on exposure to blue light, the production of four major compounds coumarin (1.26 mg/g), eclalbatin (5.00 mg/g), wedelolactone (32.54 mg/g), and demethylwedelolactone (23.67 mg/g), as well as two minor compounds β-amyrin (0.38 mg/g) and luteolin (0.39 mg/g) were increased [102].

## 3. Mechanistic Considerations and Future Prospects 

Light is a key driver of plant secondary metabolite biosynthesis and accumulation [115]. When photons activate photoreceptors, they activate signaling pathways and cause changes in gene expression. The light-absorbing properties are defined by the interaction of a photoreceptor protein and a chromophore [116]. Figure 2 depicts the absorption bands of essential photoreceptors, as well as the quantities and processes that they affect. In comparison to photosynthesis, only a small fraction of photons absorbed are used to activate photoreceptors [117]. Plants have four major photoreceptors, absorbing photons not only in the photosynthetically active radiation (PAR) domain but also in the UV and far-red regions. They interact with many signal transduction factors and are in charge of initiating various physiological changes and adaptations, including secondary metabolite production, triggered by light. 

To address secondary metabolites in plants, photosynthetically active radiation (PAR) is insufficient, and a broader wavelength range (physiologically active radiation) is required.

The protein cryptochrome absorption is comprised between the UV-A and blue light domains (340−520 nm). Phytochrome absorbs light primarily in the red/far-red area around 665 nm (Pr) and 730 nm (Pfr), but also in the blue/near-UV region. Phototropin light absorption is comprised between 350 and 520 nm, whereas UVR8 shows maximum absorption in the range between 280 and 350 nm.

Many significant players have been identified in the literature to explain how light affects secondary metabolite biosynthesis. The first is ROS; when exposed to large levels of UV-B radiation, plants preferentially produce a wide range of phenylpropanoid structures, including flavonoids, supporting the theory that flavonoids are principally engaged in photoprotection by absorbing the shortest solar wavelengths [118]. There is, however, substantial evidence that flavonoid production is similarly enhanced in response to high photosynthetically active radiation either in the presence or absence of UV radiation. For instance, the presence of phytochromes, at a modest level, as well as UVR8 and cryptochrome, are important for anthocyanin accumulation; therefore, wavelengths beyond UV also show impacts [115]. This lends credence to the idea that flavonoids can act as scavengers of ROS produced by excessive light [118].

Phytohormonal control is also an important regulation that could be linked to light. Phytohormones have been linked to the regulation of secondary metabolite synthesis under both developmental and stress circumstances [119,120]. Many phytochrome proteins have been connected to variations in phytohormone levels, including gibberellins and auxins, ethylene, jasmonates, and abscisic acid (ABA) [117,121], all of which are, for example, significant players in the control of plant secondary metabolism, including pigments [115].

Light may also regulate the expression of transcription factors, which are essential regulators of secondary metabolite biosynthesis. For instance, cryptochrome has been shown to regulate the accumulation of anthocyanins [122], whereas green light has been shown to reverse the effects of blue light in anthocyanin accumulation [123]. The expression of R2R3MYB transcription factors, including V1MYBA1-2 and VIMYBA2, which regulate the expression of anthocyanin synthesis-related genes like *VvUFGT*, increased in grape berries irradiated with blue light, which is consistent with this observation [124].

It all demonstrates how light can regulate the formation of secondary metabolites at various levels and via various signaling pathways. Yet, in vitro systems have received far too little attention at these levels of regulation, while in vitro systems could be appealing models for conducting this type of research since, unlike the whole plant, they allow for uniformity, accessibility, and reduced complexity [125].

Light allows for the manipulation of growth conditions and secondary metabolite synthesis in both controlled and in vitro cultures, but several elements must be addressed before it can be completely utilized. In recent years, LED systems appear to have a lot of promise, as light influences plant growth and development at every step. It affects morphogenesis, differentiation, and proliferation rates in the plant cell, tissue, and organ cultures, all of which are important for the biosynthesis of secondary metabolites. Conscious manipulation of light quality, along with the advantages of LED technology, will lead to increased biomass production with high secondary metabolite content under regulated in vitro settings. It will also ensure that homogenous material is obtained in a reasonably short amount of time without the need to cultivate the whole plant to the fully mature stage in the field. However, according to a review of the literature, the response appears to be depending on the plant species and the culture systems. Biosynthesis of plant secondary metabolites is regulated effectively by various signaling events, as the production of secondary metabolites depends upon the plant species and their respective phytochemical classes. As a result, proposing a general mechanism of light elicitation on the production of different secondary metabolites is complicated, and further research is needed to gain a comprehensive understanding. To uncover the molecular and cellular mechanism of light elicitation, an omics-based analysis could be a useful approach. Therefore, in conclusion, it may be inferred that different light regimes can aid in the improvement of growth and developmental alterations for the in vitro generation of safe secondary metabolites.

## 4. Conclusions

The current review deciphers the role of light as a promising elicitor in enhancing the production of key metabolites in different in vitro systems of plants. Among various abiotic elicitors, light has gained attention in enhancing secondary metabolite production due to its specified wavelengths, cost-effectiveness, and durability. A plethora of studies have reported the effect and quality of different types of light sources on enhanced secondary metabolite accumulation in several plant species in vitro. These findings pave the way for a more thorough examination of light as a plant elicitor. The fact that most studies have focused on plant growth and development, including the generation of primary metabolites, highlights the paucity of information about the different factors regulating light elicitation mechanisms. Thus, many studies still lack at depicting the mechanisms of elicitation promoting the production of pharmacologically important secondary metabolites since it may differ depending upon the type of plant species, culture conditions, and light source used. 

## Figures and Tables

**Figure 1 plants-10-01521-f001:**
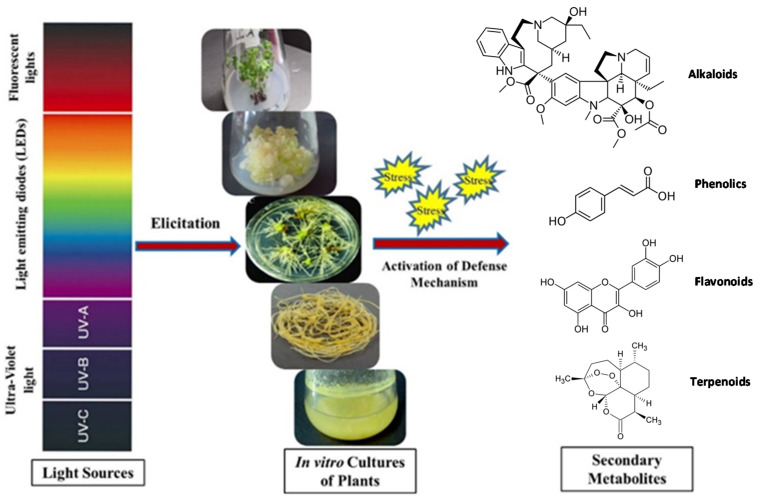
An overview of light’s function as an elicitor of important secondary metabolites in various in vitro plant cultures maintained under controlled conditions, including shoot, callus hairy root, adventitious root, and cell suspension cultures (from top to bottom). Different light sources, including UV light but also excessive light, can cause stress and activate the defense response, resulting in the production of a variety of bioactive plant secondary metabolites such as alkaloids (e.g., vinblastine), phenolics (e.g., *p*-coumaric acid), flavonoids (e.g., quercetin), or terpenoids (e.g., artemisinin).

**Figure 2 plants-10-01521-f002:**
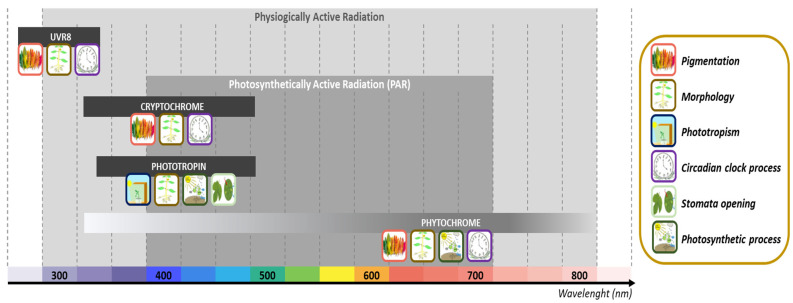
Photoreceptor absorption bands in plants and their processes (adapted from [115]). Absorbing photons activate different types of photoreceptors not just in the PAR domain but also in the UV and far-red regions. Each absorption band and photoreceptor couple are engaged in a different physiological response that may be essential for biomass and secondary metabolite synthesis, such as pigmentation, morphology, phototropism, circadian clock process, stomata opening, or photosynthetic activity.

**Table 1 plants-10-01521-t001:** Effects of UV light as an elicitor of key secondary metabolites in controlled and in vitro plant culture systems.

Light Sources	Intensity	Exposure Time	Plant Species	Culture System	Secondary Metabolite	Yield Increase	References
UV-A/B	4–5 Wm^−2^/10–14 Wm^−2^	3 h per day for 16 days	*Capsicum annum*	CGC (leaf)	Cynaroside	-	[30]
UV-B	73.08 kJ/m^2^/day	7 h per day for 13 days	*Nymphoides humboldtiana*	CGC (leaf)	Flavonoids	-	[31]
UV-B	40 J/cm^2^	8 h	*Alternanthera Sessilis*	CGC (shoot)	Flavonoids	51%	[32]
			*Alternanthera Brasiliana*			62%	
UV-B	1.14 kJ/m^2^/day	4 h per day for 14 days	*Capsicum annuum*	CGC (leaf)	Flavonoids	-	[33]
UV-B	313 nm	7–14 days	*Ginkgo biloba*	CGC (leaf)	Quercetin, kaempferol, and isorhamnetin	2.05- to 2.4-fold and 16.67- to −42-fold, respectively	[34]
UV-B	1.26 μW/cm^2^	5 min	*Catharanthus roseus*	Cell suspension culture	Catharanthine and vindoline	3-fold and 12-fold	[35]
UV-B	224 μmol m^−2^ s ^−1^	1 h for 2 days and 2 h for 2 days	*Ocimum basilicum* Green basil	CGC (leaf)	Anthocyanin, phenolics, and flavonoids	9–23%, 28–126% and 80–169%, respectively	[21]
		2 h for 2 days and 2 h for five days	*Ocimum basilicum* Purple Basil		Phenolics and flavonoids	29–63% and 37–79%	
UV-B	102 kJ/m^2^/day	3 days	*Ocimum basilicum*	CGC (plantlet)	Phenolics		[36]
UV-B	20 µW/cm^2^	4 days	*Triticum aestivium*	CGC (seedling)	Phenolics	26.3%	[37]
UV-C	254 nm	15 min	*Vitis vinifera*	Callus culture	*trans*-resveratrol	26-fold	[38]
UV-C	254 nm	5 min after 24 h	*Vitis vinifera*	Callus culture	*trans*-resveratrol	8-fold	[39]
		10 min after 48 h			Catechin	-	
UV-C	3 W/m^2^	60 min	*Lepidium sativum*	Callus culture	Chlorogenic acid, kaemferol, and quercetin	2.5-fold	[40]
UV-C + Photoperiod UV-C + Dark	3.6 kJ/m^2^ + 16/8 h 1.8 kJ/m^2^ + 24 h dark	10–60 min	*Linum usitatissimum*	Callus culture	Secoisolariciresinol diglucoside (SDG)	1.86-fold	[41]
					Lariciresinol diglucoside (LDG)	2.25-fold	
					Guaiacylglycerol-β-coniferyl alcohol ether glucoside (GGCG)	1.33-fold	
					Total phenolic production	2.82-fold	
					Total flavonoid production	2.94-fold	
					Dehydrodiconiferyl alcohol glucoside (DCG)	1.36-fold	
UV-C	3 W/m^2^	10 min	*Ocimum basilicum*	Callus culture	Rosmarinic acid	2.3-fold	[42]
					Chichoric acid and cyanide	4.1-fold	
					Peonidin	2.7-fold	
UV-C	254 nm	60 min	*Echinacea purpurea*	Callus culture	Phenolics	-	[43]
				Cell suspension culture			

CGC = controlled growth chamber.

**Table 2 plants-10-01521-t002:** Role of fluorescent light as an elicitor of key secondary metabolites in various in vitro systems of plants.

Light Type	Light Characteristic	Exposure Time	Plant Species	Culture System	Secondary Metabolite	Yield Increase	References
Blue light	380–560 nm	30 days	*Stevia rebaudiana*	Callus culture	Phenolics and Flavonoids	-	[22]
Blue light	40–50 µmol m^−2^ s^−1^	3 weeks	*Prunella vulgaris*	Callus culture	Phenolics and Flavonoids	-	[14]
Blue light	60 µmol m^−2^ s^−1^	6 weeks	*Scutellaria lateriflora* L.	Shoot culture	Glucuronides, Baicalin, Wogonoside, Verbascoside	1.54-, 1.49-, 2.05- and 1.86-fold, respectively	[72]
Red light	660 nm	5 weeks	*Hypericum perforatum*	Root culture	Hypericins, Flavonoids	-	[73]
Blue light	470 nm	1 week			Hypericins, Phenolics	52% and 26%	
Blue light	50 µmol m^−2^ s^−1^	30 days	*Hyptis marrubioides*	Micro propagation	Rutin	-	[74]
White light							
Fluores cent light	50 µmol m^−2^ s^−1^	4 weeks	*Panax ginseng*	Hairy root culture	Ginsenoside	-	[75]
Green light	40–50 µmol m^−2^ s^−1^	3 weeks	*Artemisia absinthium*	Callus culture	Phenolics and Flavonoids	-	[9]

**Table 3 plants-10-01521-t003:** Role of LED lights as an elicitor of key secondary metabolites in plant culture systems.

Light Types	Light Characteristic	Exposure Time	Plant Species	Culture System	Secondary Metabolite	Yield Increase	References
Monochromatic Blue LED	456 nm	27 days	*Solanum lycopersicum*	Closed-type plant production system (seedling)	Phenolics and flavonoids	-	[23]
Blue LED	200 µmol m^−2^ s^−1^	14/10 h	Lettuce	CGC (leaf)	Flavonoids	2.07-fold	[89]
Blue LED	50 µmol m^−2^ s^−1^	28 days	*Curculigo Orchioides*	CGC (shoot bud)	Phenolics and flavonoids	-	[90]
Blue LED	30 ±1 µmol m^−2^ s^−1^	8 h per day 40 days	*Anoectochilus roxburghii*	CGC (leaf)	Flavonoids and polyphenols	24.2%	[91]
Blue (+B) and Blue-violet (+BV) LED	450 nm and 420–440 nm	10 days	*Ocimum basilicum*	CGC (leaf)	Phenolics		[92]
			*Eruca sativa*		Flavonoids		
Red LED	35 µmol m^−2^ s^−1^	6 weeks	*Myrtus communis*	In vitro shoot culture	Gallic acid and Myricetin	-	[93]
Red LED	40–50 μmol m^−2^ s ^−1^	24 h	*Silybum marianum*	Callus culture	Silymarin	2-fold	[94]
Blue LED	50 µmol m^−2^ s^−1^	4 weeks	*Rehmannia glutinosa*	In vitro shoot culture	Phenolics and flavonoids	39.3%, 33.6%	[95]
Red LED	450 nm			CGC (root and leaf)	Phenolics, Flavonoids	33.6%, 61.7%	[96]
Blue LED	40–50 µmol m^−2^ s^−1^	4 weeks	*Ocimum basilicum*	Callus culture	Rosmarinic acid and eugenol	2.46- and 2.25-fold	[97]
Red LED					Peonidin and cyanidin	3.5- and 4.53-fold	
White LED					Chicoric acid	4.52-fold	
White LED	380 nm	4 days	*Triticum aestivium*	CGC (seedling)	Epicatechin	1.3- to 1.46-fold	[98]
Blue LED	470 nm	8 days			Gallic acid and quercetin	1.2–1.5 fold	
Red LED	660 nm	8 days			Ferulic acid	-	
		12 days			*p*-coumaric acid	1.27- to 1.77-fold	
Red Blue light	137 µmol m^−2^ s^−1^	96 h	*Fagopyrum* sp.	CGC (seedling)	Flavonoids	-	[99]
Blue LED	177 µmol m^−2^ s^−1^				Anthocyanins	-	
Red and Blue 2R:1B	120 µmol m^−2^ s^−1^	12 h each day for 17 days	*Ocimum basilicum*	CGC (seedling)	Phenolics and flavonoids	1.63–1.87-fold and 2.06-fold	[100]
Blue–red (1:1) LED	44.80 µmol/s	30 days for 16 h	*Dendrobium*	In vitro protocorn-like body culture	Flavonoids	-	[101]
Red: Blue LED	33 µmol m^−2^ s^−1^	4 weeks	*Cnidium officinale*	Callus culture	Phenolics and flavonoids	-	[10]
Red LED	660 nm	24 h	*Eclipta alba*	Callus culture	Phenolics and flavonoids	2-fold	[102]
Blue LED	460 nm						

## Data Availability

All data are included in the present study.

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
