# Peer review of "Comparative Effects of Different Light Sources on the Production of Key Secondary Metabolites in Plants In Vitro Cultures"

_plants, 2021, doi:10.3390/plants10081521_

Round 1

Reviewer 1 Report

  1. Figure 1: Triterpenoids structure should be resized: cause of wrong proportion
  2. Line 99: too long a gap between words
  3. Lines 107-110: This source (citation “29” Ahmad et al. 2019) in my opinion not treated about the elicitation. Mentioned compounds was extracted with organic solvents, but UV was used as a factor that diminished biological activities of these extracts. Probably through structure degradation of compounds responsible for their activities. I suggest to delete, because it is confusing to the readers.
  4. Line 144: “virulent metabolites”: what the term is about?
  5. Line 180: should be 20 μM, not M
  6. Line 190: one compound name (Secoisolariciresinol) was written by capital letter, but other not…?
  7. Line 197: should be 3.6 kJ/m2
  8. Line 288, Tab.2 and line 422, Tab.3: kolumn “Intensity” contains in few cases wavelength. I suggest change “Intensity” to “light parameters” or “light characteristic”
  9. Line 314: “reflection” should not be “reflectance”?
  10. Line 382: „±” and „μ” were absent, should be „35 ± 0.05 μg GAE/mg”
  11. Line 410: should be “mmol/g FW”

Author Response

AUTHORS: Thank you very much for your very useful comments, we have revised our manuscript accordingly.
1. Figure 1: Triterpenoids structure should be resized: cause of wrong proportion
AUTHORS: Figure 1 has been modified accordingly.
2. Line 99: too long a gap between words
AUTHORS: Abstract has been reduced to 182 words.
3. Lines 107-110: This source (citation “29” Ahmad et al. 2019) in my opinion not treated about the elicitation. Mentioned compounds was extracted with organic solvents, but UV was used as a factor that diminished biological activities of these extracts. Probably through structure degradation of compounds responsible for their activities. I suggest to delete, because it is confusing to the readers.
AUTHORS: Thank you for this remark. We deleted this reference accordingly.
4. Line 144: “virulent metabolites”: what the term is about?
AUTHORS: We changed it to “bioactive metabolites”.
5. Line 180: should be 20 μM, not M
AUTHORS: Thank you. Correction done.
6. Line 190: one compound name (Secoisolariciresinol) was written by capital letter, but other not…?
AUTHORS: Thank you. Correction done.
7. Line 197: should be 3.6 kJ/m2
AUTHORS: Thank you. Correction done.
8. Line 288, Tab.2 and line 422, Tab.3: kolumn “Intensity” contains in few cases wavelength. I suggest change “Intensity” to “light parameters” or “light characteristic”
AUTHORS: Thank you. Correction done.
9. Line 314: “reflection” should not be “reflectance”?
AUTHORS: Thank you. Correction done.
10. Line 382: „±” and „μ” were absent, should be „35 ± 0.05 μg GAE/mg”
AUTHORS: Thank you. Correction done.
11. Line 410: should be “mmol/g FW”
AUTHORS: Thank you. Correction done.

Reviewer 2 Report

I appreciate author's effort during the construction of the manuscript. 

I will suggest authors to add future prospects before conclusion. 

Please see attached pdf for more corrections. 

Author Response

REVIEWER 2:

I appreciate author's effort during the construction of the manuscript. 

AUTHORS: Thank you very much for your very useful comments, we have revised our manuscript accordingly.

I will suggest authors to add future prospects before conclusion. 

AUTHORS: We add a paragraph with future prospects before conclusion.

Please see attached pdf for more corrections. 

AUTHORS: Thank you. Corrections done.

Reviewer 3 Report

The article is very interesting. Can be published.

The article reports the interesting experimentation and correlation between the use of different types of illumination  ( in vitro experimentation) and the production of secondary metabolites by different crops. The topic is really interesting and original as it is trying to use more and more natural products instead of synthetic ones, in the agricultural and the therapeutic field.
The article is written in a clear understandable but above all captivating way
The conclusions are consistent with the text. Obviously, the in vitro experimentation must subsequently be conducted in vivo. It is also not possible to generalize the results as these are related to the type of crop.

Author Response

REVIEWER 3:

The article is very interesting. Can be published.

AUTHORS: Thank you very much for your very useful comments, we have revised our manuscript accordingly.

The article reports the interesting experimentation and correlation between the use of different types of illumination  ( in vitro experimentation) and the production of secondary metabolites by different crops. The topic is really interesting and original as it is trying to use more and more natural products instead of synthetic ones, in the agricultural and the therapeutic field.
The article is written in a clear understandable but above all captivating way
The conclusions are consistent with the text. Obviously, the in vitro experimentation must subsequently be conducted in vivo. It is also not possible to generalize the results as these are related to the type of crop.

AUTHORS: We add a paragraph mentioning these aspects with future prospects before conclusion.

Reviewer 4 Report

The paper entitled 'Comparative Effects of Different Light Sources on the Production of Key Secondary Metabolites in Plants In Vitro Cultures' is based on characterization of various type of light sources on secondary metabolites in plants. On the hole, the paper is well prepared. Authors presented the subject in detailed. References are up-to-date, necessary information are presentd in form of table. The paper in suitable for publication in Plants neverheless some points should be improved:

  1. Abstract is too long. This part should include max. 200 words.
  2. All tables should be improved because the information are hard to read
  3. Sections UVA, UVA/UVB and UVB should be enriched in additional information. Please improve these sections.

Author Response

REVIEWER 4:

The paper entitled 'Comparative Effects of Different Light Sources on the Production of Key Secondary Metabolites in Plants In Vitro Cultures' is based on characterization of various type of light sources on secondary metabolites in plants. On the hole, the paper is well prepared. Authors presented the subject in detailed. References are up-to-date, necessary information are presentd in form of table. The paper in suitable for publication in Plants neverheless some points should be improved:

AUTHORS: Thank you very much for your very useful comments, we have revised our manuscript accordingly.

  1. Abstract is too long. This part should include max. 200 words.

AUTHORS: Abstract has been reduced to 182 words.

  1. All tables should be improved because the information are hard to read

AUTHORS: Tables have been revised accordingly.

  1. Sections UVA, UVA/UVB and UVB should be enriched in additional information. Please improve these sections.